# Effects of Thiamethoxam-Dressed Oilseed Rape Seeds and *Nosema ceranae* on Colonies of *Apis mellifera iberiensis*, L. under Field Conditions of Central Spain. Is Hormesis Playing a Role?

**DOI:** 10.3390/insects13040371

**Published:** 2022-04-09

**Authors:** Elena Alonso-Prados, Amelia Virginia González-Porto, Carlos García-Villarubia, José Antonio López-Pérez, Silvia Valverde, José Bernal, Raquel Martín-Hernández, Mariano Higes

**Affiliations:** 1Unidad de Productos Fitosanitarios, Instituto Nacional de Investigación y Tecnología Agraria y Alimentaria (INIA, CSIC), 28040 Madrid, Spain; 2Laboratorio de Mieles y Productos de las Colmenas, Centro de Investigación Apícola y Agroambiental (CIAPA), Instituto Regional de Investigación y Desarrollo Agroalimentario y Forestal (IRIAF), Consejería de Agricultura de la Junta de Comunidades de Castilla-La Mancha, 19180 Marchamalo, Spain; avgonzalezp@jccm.es; 3Experimentación e Investigación Hortícola Centro de Investigación Apícola y Agroambiental, Centro de Investigación Apícola y Agroambiental (CIAPA), Instituto Regional de Investigación y Desarrollo Agroalimentario y Forestal (IRIAF), Consejería de Agricultura de la Junta de Comunidades de Castilla-La Mancha, 19180 Marchamalo, Spain; cgarciav@jccm.es (C.G.-V.); jalopezp@jccm.es (J.A.L.-P.); 4Instituto Universitario CINQUIMA, Analytical Chemistry Group, Universidad de Valladolid, 47011 Valladolid, Spain; silviaval2308@gmail.com (S.V.); jose.bernal@uva.es (J.B.); 5Instituto de Recursos Humanos para la Ciencia y la Tecnología (INCRECYT–ESF/EC-FSE), Fundación Parque Científico y Tecnológico de Castilla–La Mancha, 02006 Albacete, Spain; rmhernandez@jccm.es; 6Laboratorio de Patología Apícola, Centro de Investigación Apícola y Agroambiental (CIAPA), Instituto Regional de Investigación y Desarrollo Agroalimentario y Forestal (IRIAF), Consejería de Agricultura de la Junta de Comunidades de Castilla-La Mancha, 19180 Marchamalo, Spain; mhiges@jccm.es

**Keywords:** honey bees, Spain, thiamethoxam, *Nosema ceranae*, hormesis, *Brassica napus*, oilseed rape, seed treatment, EPPO, viruses

## Abstract

**Simple Summary:**

The collapse of the honey bee colonies is a complex phenomenon in which different factors may participate in an interrelated manner (e.g., pathogen interactions, exposure to chemicals, beekeeping practices, climatology, etc.). In light of the current debate regarding the interpretation of field and monitoring studies in prospective risk assessments, here we studied how exposure to thiamethoxam affects honey bee colonies in Central Spain when applied as a seed treatment to winter oilseed rape, according to the good agricultural practice in place prior to the EU restrictions. Under the experimental conditions, exposure to thiamethoxam, alone or in combination with other stressors, did not generate and maintain sufficient chronic stress as to provoke honey bee colony collapse. The stress derived from exposure to thiamethoxam and honey bee pathogens was compensated by adjustments in the colony’s dynamics, and by an increase in the worker bee population, a behavior known as hormesis. An analysis of the factors underlying this phenomenon should be incorporated into the prospective risk assessment of plant protection products in order to improve the future interpretation of field studies and management practices.

**Abstract:**

To study the influence of thiamethoxam exposure on colony strength and pathogen prevalence, an apiary (5 colonies) was placed in front of a plot sown with winter oilseed rape (wOSR), just before the flowering phase. Before sowing, the seeds were treated with an equivalent application of 18 g thiamethoxam/ha. For comparison, a second apiary (5 colonies) was located in front of a separate 750 m plot sown with untreated wOSR. Dead foragers at the entrance of hives were assessed every 2–3 days throughout the exposure period, while the colony strength (number of combs covered with adult honey bees and brood) and pathogens were monitored each month until the following spring. Foraging on the wOSR crop was confirmed by melissopalynology determination of the corbicular pollen collected periodically, while the chemical analysis showed that exposure to thiamethoxam was mainly through nectar. There was an increase in the accumulation of dead bees in the apiary exposed to thiamethoxam relating with the control, which was coped with an increment of bee brood surface and adult bee population. However, we did not find statistically significant differences between apiaries (α = 0.05) in terms of the evolution of pathogens. We discuss these results under hormesis perspective.

## 1. Introduction

In the past decade, significant losses of honey bee colonies have been reported, mainly in Europe and North America [1,2,3,4,5,6,7,8,9,10,11]. The consensus is that several factors are involved in this phenomenon: nutritional stress, pathogens, hive management practices, exposure to multiple xenobiotic residues (pesticides and veterinary drugs), invasive species, seasonal weather changes and genetic variability [7,12,13,14,15,16,17,18,19]. However, exposure to chemicals and the most prevalent pathogens are without doubt the main drivers in the phenomenon of honey bee colony loss [20].

Among the most relevant chemicals, much attention has been paid to neonicotinoids [21] in recent years due to their widespread use [22,23,24,25]. These pesticides selectively act on the insect’s central nervous system (CNS) as they are agonists of post-synaptic acetylcholine receptors [26,27,28,29]. Neonicotinoids have a low molecular weight, they are moderately-to-highly water soluble and they have a low octanol:water partitioning coefficient. These physico-chemical properties favor the efficient translocation of these chemicals through the plant xylem, such that they can systemically control piercing-sucking insects [24]. However, these characteristics also mean they have a high capacity for environmental transport [30,31,32]. Depending on the toxophore, these compounds are classified as N-nitroguanidines (imidacloprid, thiamethoxam, clothianidin and dinotefuran), nitromethylenes (nitenpyram), or N-cyano-amidines (acetamiprid and thiacloprid) neonicotinoids, with the former representing the class that has produced most concern for bees [33]. Many sub-lethal effects of N-nitroguanidine neonicotinoids have been identified in bees, influencing on orientation and olfactory learning, flight, queen performance, honey bee colony physiology and the bee’s immune system [34,35,36,37,38,39,40,41,42,43,44,45,46,47]. In particular, thiamethoxam (TMX) causes damage in the midgut, brain cells and Malpighian tubules of bees [48,49,50], affecting several biological process such as oxidative phosphorylation and tyrosine metabolism [51], as well as altering the gut microflora [52].

On the other hand, *Nosema ceranae* and *Varroa destructor* have been intensively studied in relation to the phenomenon of honey bee colony losses. *N. ceranae* is found worldwide [14,53,54], it has a high pathogenic capacity [55,56] and it is transmitted horizontally in the colony, through food, water, trophallaxis and cleaning duties (reviewed in [54]). It is an obligate intracellular parasite that depends on the host’s energy for reproduction, and it alters the physiology and behavior of its host to ensure conditions are optimal for it to complete its biological cycle (reviewed in [54]). As such, *Apis mellifera* individuals infected by *N. ceranae* experience: (1) a strong carbohydrate demand; (2) altered lipid and amino acid metabolism; (3) an altered vitellogenin/juvenile hormone equilibrium; (4) a reduction in hexamerins and major royal jelly proteins; (5) an upregulation of the octampamide pathway; (6) a suppressed immune response; (7) impaired apoptosis of ventricular epithelial cells [54,57,58], and (8) changes in midgut microbiota [52]. As a consequence of all these alterations, *N. ceranae* will perturb colony homeostasis if the infection is maintained over time [59]. In this way, infected bees generate higher Ethyl Olate (EO) titers and undertake more flight activity than non-infected bees [60]. The constant reduction in bee number due to the tissue damage, energetic stress, and altered flight behavior due to infection accelerates age polyethism in young bees to cover the energy demands of the colony [61,62]. However, precocious foragers (FBs) are less effective, so higher number of FBs will be needed to fulfil the colony’s demands, reducing the time each bee can dedicate to hive tasks. Thus, if the buffering capacity of the colony is exceeded then there is a strong risk of collapse [54].

For its part, *V. destructor*, one of the most important pathogens of honey bees [63,64,65], not only damages the bee by feeding on its hemolymph and fat body [66,67] but also, it acts as a vector for a number of viruses [68]. As a result, this pathogen impairs the responses of the immune system [69] and affects the nervous system [70,71].

Honey bees exposed to neonicotinoids may be more susceptible to *N. ceranae* and *V. destructor* infections and that of other viruses [72]. Indeed, *N. ceranae*-infected honey bees suffer increased mortality under laboratory conditions when exposed to sub-lethal concentrations of imidacloprid [73,74], thiacloprid [75,76] or TMX [77]. Similarly, the damage caused by *N. ceranae* in midgut cells [78] may make infected honey bees more susceptible to neonicotinoid exposure. The negative synergistic effects of *Varroa* and neonicotinoid insecticides have been related to immune suppression, which impairs antiviral immune barriers [47] and memory retention [79].

However, as all these effects are rarely observed under real world conditions [80,81], they remain somewhat controversial.

Here, we present the results of a field study undertaken in accordance with good beekeeping practice (prior to European restrictions) in Central Spain that set out to determine the evolution of honey bee colonies and the prevalence of pathogens following exposure to TMX through the pollen and nectar of seed-treated oilseed rape. The stress provoked by both TMX and pathogens was compensated by the dynamics of the colonies, mainly through an increase in the bee population. We discuss the possible reasons for these responses from the hormesis point of view and in the context of the prospective risk assessment of plant protection protocols.

## 2. Materials and Methods

### 2.1. Experimental Design

The experiments were designed in accordance with the EPPO 170 standards [82]. Colonies of *Apis mellifera iberiensis* from the same apiary located at the ‘Centro de Investigación Apícola y Agroambiental” (CIAPA, Marchamalo, Guadalajara, Spain) and with naturally mated sister queens of the same age, were used to monitor the presence of queens, and to analyze and control the sanitary status of the colonies in order to ensure the colony’s strength at the beginning of the study (10 brood chamber combs covered by bees). The experimental colonies were randomly divided into two groups of five colonies each, and they were placed in front of two plots of 2 ha (750 m apart), that were sown with winter oilseed rape (wOSR, Ginfizz variety), just before their flowering phase (Figure 1).

Sowing took place on 29 October 2014 at a density of 75 seeds m^−2^. In one plot, the seeds were treated with Cruiser 350 FS© before sowing (TMX 35% p/v FS, 1200 mL/100 kg seed), while both fields were treated with Butisan S© (metazachlor 50% p/v SC) at 1.5 L/ha on 31 November, receiving no further phytosanitary treatments during the study. The fields were not treated with plant protection products for at least 2 years and before sowing, the soil of the plots was sampled and characterized as described in Section 2.2.

The exposure phase lasted from 8 April to 11 May 2015, at the beginning of which an upper box was placed in each hive, which contained empty combs ready to be used by the honey bees. The stored pollen (beebread) and honey in the brood chambers was removed to stimulate foraging activity during the exposure period, and to ensure that the pollen and nectar came from the experimental plots [83]. After the exposure phase, the colonies were placed 35 km away from the area of exposure in order to monitor them until the following spring (Figure 1).

To avoid winter collapse due to nosemosis C, the colonies were treated exceptionally with Fumidil B on 17 September 2015 under controlled conditions. The treatment consisted of 4 doses of 30 mg fumagillin per colony dissolved in 250 cc of sugar syrup (50% sugar/distilled water), administered in plastic bags placed over the brood at one-week intervals to ensure full consumption. This posology was effective without leaving residues in the honey [84]. *V. destructor* was controlled at the beginning of September with Apitraz© (amitraz a.m.), employed at the recommended dose and in accordance with current Spanish legislation [85].

### 2.2. Soil Characterization and Climatological Data

The soil surface layer (0–20 cm) was characterized following ISRIC protocols [86] and using three samples gathered randomly in each plot. The anion analysis (F^−^, Cl^−^, NO_2_^−^, NO_3_^−^, PO_4_^3−^, SO_4_^2−^) was carried out at room temperature using a Metrohm Compact IC model 761 ion chromatograph (Metrohm, Herisau, Switzerland). Na^+^ and K^+^ were measured using a flame photometry SHERWOOD 410 apparatus, and Ca^2+^ and Mg^2+^ using atomic absorption spectrophotometry (ANALYTIKJENA NOVAA 300). Electrical conductivity (EC), pH, and humidity were measured with an EC meter CRISON micro CM2000, a pH meter CRISON GLP 21, and a thermogravimetric balance KERN DBS, respectively. Bouyoucos’ densitometry method was followed to characterize the soil texture [87], and the organic matter and carbonate content was determined with the Walkley–Black and acid neutralization methods [86], respectively.

A Walter–Lieth diagram [88] was developed with historical weather data obtained from the meteorological station at Marchamalo from the regional Irrigation Advisory Service (SIAR-CREA) network to compare it with data during the period of study. These diagrams illustrate the rainfall and temperature changes throughout the year in standardized charts which provide brief summaries of the average climatic variables and their time course. The diagrams were drawn up with the diagnostic tool of the Worldwide Bioclimatic Classification System, 1996–2021 [89].

### 2.3. Sampling Schedule

The number of dead worker bees was counted in box dead-bee traps [90] at the entrance of hives every 2–3 days throughout the exposure period. Colony strength was checked by determining the number of combs covered by brood and bees once every two months [59]. The honey production in each colony was evaluated in the harvesting season as the difference in the weight of the combs before and after honey extraction [91].

Corbicular pollen loads were collected using standard traps, activated for 24 h at the beginning (8 April), in the middle (15 April), and at the end of blooming (29 April). For melissopalynological and chemical analyses, at the end of the exposure period, beebread and honey samples were collected individually from one frame of the brood chambers and the upper boxes, respectively. Beebread was extracted aseptically, removing the wax from the combs as described previously [92,93,94], and storing the samples at −80 °C in the dark until they were analyzed. The honey was sampled using a sterile spatula and stored at −20 °C.

To identify viruses and quantify the *Nosema* spp. parasitic load in the colonies, both non-foragers, (IBs, bees sampled from no-brood combs, *n* > 60), and foragers (FBs, bees arriving at the colony after closing the hive entrance for 30 min, *n* > 60) were collected separately from each colony (Table 1) and frozen (−20 °C).

### 2.4. Palynological and Melissopalynological Assessment

To confirm the type of foraging flora, corbicular pollen, beebread and honey samples were analyzed according to previously described methods [95,96]. Corbicular pollen samples were weighed and stored at −20 ± 2 °C until further analysis. Pollen loads were separated based on color and texture to identify and determine the contribution of each botanical type into the samples. The pollen grains were isolated from each sample and cleaned using the Erdtam method [97]. One aliquot from each of the ten pollen loads of the same “color-texture” class was placed onto a slide in glycerin jelly mounting medium and examined microscopically in order to ensure the homogeneity of identification. The proportion of each pollen type was calculated by dividing the weight obtained for each color fraction by the overall weight of the sample analyzed [95].

Pollen was extracted from the beebread by diluting 0.5 g in 10 mL of 0.5% acidulated water (96% sulfuric acid) and centrifuging at 2500 r.p.m. for 15 min. The pellet was washed with double-distilled water and centrifuged twice, and the sediment was placed onto a slide in glycerin jelly mounting medium and examined microscopically to identify the pollen. The frequency of Brassica type pollen grains (*Brassica* t. pollen) was expressed as a percentage of the total pollen grains.

The honey samples were treated chemically with 0.5% acidified water (sulfuric acid 96%), and a qualitative and quantitative analysis was performed on the sediment recovered from 5 g aliquots [98]. Between 300 and 1200 pollen grains were counted in each sample, and the pollen grains were identified and classified on the basis of the identification keys [99,100]. All the reference collection pollen slides used were available at the honey laboratory at the CIAPA.

### 2.5. Chemical Analysis

Chemical analyses were carried out as described elsewhere [101,102,103]. FLUKA-PESTANAL analytical standards of clothianidin (CLO; purity 99.9%), TMX (purity 99.6%), and TMX-d3 (isotope-labeled standard; purity ≥ 98.0%) were purchased from Sigma-Aldrich Laborchemikalien (Seelze, Germany). LC grade methanol (MeOH), and acetonitrile (ACN) were supplied by Lab Scan Ltd. (Dublin, Ireland). Formic acid (FA), ammonium formate, and ethyl acetate (EA) were obtained from Sigma-Aldrich Chemie Gbmh (Steinheim, Germany); sodium chloride, trisodium citrate dihydrate, and trisodium citrate sesquihydrate were supplied by Panreac (Barcelona, Spain). Meanwhile, primary secondary amine (PSA) and octadecylsilane (C_18_) were provided by Supelco (Bellefonte, PA, USA). Finally, ultrapure water was obtained using Milipore Mili-RO plus and Mili-Q systems (Bedford, MA, USA). All chemicals used were of analytical grade.

Briefly, to quantify the levels of TMX and its metabolite clothianidin (CLO) in the seeds [101], insecticides were extracted with a mixture of acetonitrile (ACN) and sodium chloride (60:40, *v*/*v*). After centrifugation (5810 R refrigerated bench-top Eppendorf centrifuge; Hamburg, Germany), the supernatant was collected and concentrated at 60 °C (R-210/215 rotary evaporator from Buchi, Flawil, Switzerland). The dry extract was reconstituted with 1 mL of a mixture of 0.1% (*v*/*v*) FA in ACN and 0.1% (*v*/*v*) FA in water (25:75, *v*/*v*). The extract was filtered and injected onto a liquid chromatography coupled to a diode array detector (DAD; Agilent Technologies 1200 series; Palo Alto, Santa Clara, CA, USA) column (Kinetex C_18_, 150 × 4.6 mm, 2.6 μm, 100 Å; Phenomenex, Torrance, CA, USA) system. The limits of quantifications (LOQs) were set at 0.11 and 0.15 g/kg for TMX and CLO, respectively [101].

In relation to the analysis in bee pollen, it was employed as a sample treatment modified QuEChERS (quick, easy, cheap, effective, rugged, and safe) method. A representative amount of sample (1.0 g) was mixed with 2 mL of water and 6 mL of ACN, and the resulting mixture was shaken. Then, magnesium sulfate (1.0 g), sodium chloride (0.5 g), and tri-sodium citrate dihydrate (0.8 g) was added to the mixture, and after centrifugation (10,000 rpm, 10 °C, 5 min), 2 mL of the supernatant was transferred to an Eppendorf tube. After that, magnesium sulfate (150 mg), PSA (25 mg), and C_18_ (25 mg) were added to the tube, which was again centrifuged. Supernatant (1 mL) was collected and evaporated to dryness (60 °C). A further reconstitution of the dry extract was performed with 1 mL of a MeOH and water (80:20, *v*/*v*) mixture, and the resulting extract was filtered and analyzed by using an ultra-high performance liquid chromatography (UHPLC; ACQUITY, Waters, Milford, MA, USA) coupled to quadrupole time-of-flight mass spectrometry (qTOF; maXis impact, Bruker Daltonik GmbH, Bremen, Germany) system equipped with an electrospray interface (ESI), which was operated in positive mode. The obtained LOQs were 2.1 ng/g for TMX and 3.9 ng/g for CLO [102].

Finally, two different sample treatments were employed when determining TMX and CLO depending on the botanical origin of honey. Insecticides were extracted from light color honeys (multifloral and rosemary) by using a modified QuEChERS protocol. Briefly, 5.0 g of honey was mixed with 10 mL of water and 10 mL of an ACN and EA (70:30, *v*/*v*) mixture, and after shaking, different amounts of magnesium sulfate (2.0 g), sodium acetate (1.0 g), trisodium citrate dihydrate (1.5 g), and trisodium citrate sesquihydrate were added. Then, the mixture was shaken in an ultrasound device (J.P. Selecta S.A., Barcelona, Spain), centrifuged (10,000 rpm, 10 °C, 5 min); the supernatant was collected and evaporated to dryness (60 °C), while the dry extract was reconstituted with 1 mL of a MeOH and water (80:20, *v*/*v*) mixture, being the resulting extract filtered and analyzed by an ACQUITY UHPLC (Waters) coupled to a Xevo TQ-S (triple quadrupole, QqQ) mass spectrometer (Waters) equipped with an ESI interface (positive mode). The obtained LOQs were 0.06 ng/g for TMX and 0.20 ng/g for CLO [102]. On the other hand, dark honeys (heather) required a solid-phase extraction (SPE) procedure. A representative amount of dark honey (5.0 g) was diluted in 10 mL of ammonium formate (10 mmol/L) in water. The resulting mixture was loaded onto a polymeric SPE cartridge (Strata^®^ X; Phenomenex), which was previously conditioned with 5 mL of MeOH and 5 mL of water. Then, after 5 min of drying time, the insecticides were eluted from the cartridge with 4 mL of an ACN and EA (80:20, *v*/*v*) mixture. The extract was evaporated to dryness, and the dry residue reconstituted with 1 mL of a MeOH and water (80:20, *v*/*v*) mixture, filtered and analyzed by using the same UHPLC-QqQ system that was employed for analyzing light honeys. The LOQs were 0.06 ng/g for TMX and 0.20 ng/g for CLO [103].

### 2.6. Identification of Pathogens

#### 2.6.1. Detection of Varroa Mites

At each sampling time, a total of 60 FBs and 60 IBs were examined individually per colony, using sterile tweezers to detect malformations and collect parasitic mites to be identified macroscopically. A honey bee colony was considered to be infested with *V. destructor* when at least 1 *Varroa* mite was found in the sample. The rate of infestation of the bee colony was estimated by assessing the number of *Varroa* mites relative to the number of adult bees in each sample, and it was expressed as the number of *Varroa* mites/100 bees/sample.

#### 2.6.2. Pathogen Screening

DNA and RNA were extracted at each sampling time (see Table 1) as described previously [104]. In summary, the abdomen was removed from 30 FB and 30 IB bees in each colony, placing them individually in a well of a 96-well plate containing glass beads (2 mm: Sigma^®^, St. Louis, MO, USA). The tissue was macerated in AL buffer (Qiagen^®^ 19075, Hilden, Germany) and incubated with proteinase K under the same conditions as described in [104]. DNA and RNA purification were performed simultaneously in a Biosprint 96 workstation (Quiagen^®^, Hilden, Germany) according to the BS96 DNA Tissue extraction protocol. Total nucleic acids (RNA and DNA) were eluted in 100 µL of elution buffer.

An aliquot of 75 µL was used to detect *Nosema* spp. according to an earlier procedure [105]. To identify *N. ceranae* and *N. apis*, an internal control (IPC) based on mitochondrial cytochrome oxidase subunit I (COI) was used as a target in the triplex PCR assay [105]. The percentage of FBs or IBs parasitized by Nosema was estimated as 100 × number of positives/30.

The remaining 25 µL aliquots of the total nucleic acids eluted from each FB and IB sample from each colony and sampling time were pooled immediately and subjected to DNase I digestion to remove DNA (Quiagen^®^ kit 79254), as described previously [104]. The total RNA recovered was used immediately to generate first strand cDNA with the iScriptTM cDNA Synthesis Kit (Biorad^®^, Hercules, CA, USA). The resultant cDNA was used to analyze the viruses present. Black Queen Cell Virus (BQCV) and Deformed Wings Virus (DWV) were detected by Real Time-PCR as detailed in [104], and using the primers and probes described by [106]. The Acute Bee Paralysis Virus-Kashmir Bee Virus-Israeli Acute Paralysis Virus (AKI) complex was detected by following the procedure described in [107]. In all cases, negative and positive controls were run in parallel for each step (sample processing, nucleic acid extraction, reverse transcription, and RT-PCR and PCR analysis).

### 2.7. Statistical Analysis

The statistical analysis was made according to the recommendations in [108,109]. Colony strength or pathogen parameters were compared between the two groups of colonies at the end of the exposure period (*n*(control) = *n*(TMX) = 5) with a one tailed Mann–Whitney U test (exact approach, alpha = 0.05). The Mann–Whitney U test compares the distribution of the ranks of the two populations, and to interpret this as a comparison of medians, the distributions of the two populations are assumed to be the same shape [110]. Moreover, we reported the effect size for non-parametric tests following the approach given elsewhere [111]. Mann–Whitney U tests and effect sizes were generated using the Real Statistics Resource Pack software [112] and transformed to the Cohen’s d coefficient as described previously [113]. Finally, we used G-Power 3.1.9.6 [114] to estimate the power (1-β) of the statistical tests.

To explore the evolution of the percentage of *Nosema* spp. infection in each honey bee colony, we used a Friedman test. This test is the non-parametric alternative to one-way ANOVA with repeated measures [115] and it was implemented with Statgraphics Centurion 18.

## 3. Results

### 3.1. Soil Characterization and Climatic Conditions

The 0–20 cm soil horizons of the control and treatment plots were classified as slightly acid clay and neutral clay loam, respectively, in both cases with low organic matter content and soil conductivity. The chemical characterization of the soil showed the CEC is dominated by Ca^2+^ (Appendix A). The soils are derived from Miocene-tertiary aged fluvio-alluvial sediments from Henares River Basin [116], and following the classification criteria of the World Reference Base (WRB) for Soil Resources [117], they could tentatively be classified as calcic luvisols.

Based on 12 years of climatological data (period 2010–2021) from the nearest station to the experimental plots (Appendix A), and on the bioclimatic indices estimated with the diagnostic tool of the Worldwide Bioclimatic Classification System [89], the climate in the experimental plots falls into the Upper Mesomediterranean Low Dry Blioclimatic belt, with a freezing period extending from December to March. During the period of exposure, the mean temperature was 13.0 ± 2.0 °C in April and 17.8 ± 2.5 °C in May 2015. The relative humidity ranged from 38.8 to 92.9 (mean = 61.58 ± 13.33) and the total rainfall was 29.3 mm, concentrated in 9 days and with 52.56% of the precipitation falling on 26 April 2015.

When compared to historic average climate data (Appendix A), the Walter–Lieth diagrams show that autumn 2014 (Appendix A) was wetter than the mean values for the period between 2010 and 2021. However, 2015 was characterized as being dryer and hotter than average, and with a wider range of temperatures (Appendix A). By contrast, spring 2016 was more temperate and significantly wetter, especially in April (Appendix A).

### 3.2. Palynological and Melissopalynological Assessments

The *Brassica* t. pollen content varied in the corbicular pollen sampled in April (Figure 2a). In addition to *Brassica* t. pollen, *Prunus* t. and *Cistus* t. pollens were also predominant in the samples at the beginning of the month. These taxa are typical of regression succession steps of the chorological series *Bupleuro rigidi-Querceto rotundifoliae sigmetum* in the Castellano-Maestrazgo-Manchega province [118]. In the middle of April, *Brassica* t. pollen was replaced by pollen from the surroundings, mainly *Cistus* t pollen, which was in turn replaced by *Brassica* t. pollen at the end of the month. At the end of the exposure period, the *Brassica* t. pollen content in the beebread was above 50% in all the colonies except two (C-01 and C-08), and the *Brassica* t. content of the honey in the cells of the frame introduced at the beginning of the exposure period ranged from 50.3% to 94.2% (Figure 2b).

### 3.3. Chemical Analysis

The chemical analyses revealed the level of TMX residues in treated seeds was 4660 mg/kg seed, equivalent to an application of 18 g TMX/ha. The TMX residues in bee pollen (corbicular pollen) were below the LOQ in all samples, while in the honey samples, it varied between below the limit of detection (<LOD) and 144 µg/kg (Table S13 in [103]).

### 3.4. Field Monitoring of the Honey Bee Colonies

During the exposure period, the accumulated honey bee death was higher in the treatment apiary (TMX) as the combs became covered with worker honey bees and brood (Figure 2, Table 2). Subsequently, both the worker honey bee population and the brood balanced out in both apiaries (Figure 3).

The mean number of dead bees was 127.6 ± 44.1 and 214.4 ± 116.5 in the control and treatment (TMX) apiaries, respectively. Likewise, the average number of combs covered with adult honey bees and honey bee brood was 12.2 ± 1.095 and 9 ± 4.96 in the controls, respectively, and 15.2 ± 2.05 and 11 ± 1.82 in the treatment apiaries. The same trend was observed in terms of production over the period of exposure, reaching 24.75 ± 15.05 (control) and 31.98 ± 12.21 kg (TMX) at the end of the exposure phase, respectively (Table 2).

Due to the high dispersion of the data within the apiaries (Table 2), at the end of the exposure period, a one-tailed Mann–Whitney U test did not find significant differences in the dead bees accumulated (Mann–Whitney U = 6; *p*-value = 0.111), the number of combs with bee brood (U = 6.5; *p*-value = 0.343) or in the honey produced (U = 9; *p*-value = 0.274). As such, the statistical power varied between 0.183 and 0.397 (Table 2). Regarding the number of combs covered with bees, the test did reveal a significant difference (U = 4.5; *p*-value = 0.048; 1-β = 0.72).

### 3.5. Identification of Pathogens

While *N. apis* was not detected in the colonies at any point in the study, the percentage of infection of *N. ceranae* increased in both non-forager (IB) and forager bee (FB) populations until fumagillin treatment was applied (Figure 4). The percentage of infection was significantly different between the distinct honey bee populations (Friedman Test statistic = 71.4394; *p*-value = 1.132 10–14) but not between apiaries. The maximum percentage infection was observed at the end of summer, with a mean of 62.6 ± 10.6% of FBs infected and a mean of 17.6 ± 5.06% of IBs.

*Varroa* mite was not detected in any honey bee sampled and DWV was found at a low frequency until the end of the summer, yet it was detected in all the colonies just before wintering (Table 3). By contrast, BQCV was detected throughout the experiment (Table 3), whereas the AKI complex was not identified in any sample.

## 4. Discussion

The foraging activity of honey bees in the experimental fields was guaranteed during the period of exposure due to the melliferous potential of the oilseed rape crop. Its attractiveness to honey bees is derived from the features of its flowers, their fragrance and nutritive nectar, and the high flower density of its plants, opening successively for 3–4 weeks [119]. The amount of *Brassica* t. in the bee pollen varied throughout the month of April, decreasing in the middle of April when the bulk flowering of other species took place, such as the plants of the genus *Cistus* [120,121], a producer of high quality pollen [122,123]. The chemical analysis confirmed that the honey bees were exposed to TMX through nectar rather than pollen [103]. The levels of residues in wOSR plants may have been influenced by the time between the sowing (Autumn 2014) and the exposure time (Spring 2015), and possibly due to soil leaching. Given their physico-chemical properties [24], TMX and its metabolite chlothianidin have high potential for lixiviation. This fact, together with the precipitation that fell in the days following drilling and the low organic matter content of the soils, may have favored the low retention of the residues of these chemicals in soils.

Several detections above LOQ were found in the honey samples of the control apiary [103], however, this does not invalidate the results. Thus, an increment on the mean accumulated deaths in the apiary located in front of the treated plot was observed when compared with the control at the end of the exposure period. Taking into account that exposure during foraging is not a discrete process but distributional [124], the mean colony exposure remained low even though effects were seen in a number of bees. The loss of forager bees can accelerate behavioral changes aimed at increasing food gathering in order to maintain the energy balance within the colony [125]. These changes bring about an increase in bee brood to compensate the interactions within the colony [126], referred to as a hormetic-like effect [127,128,129,130]. Hormesis is common in arthropods [131,132] and it is a dose-dependent phenomenon where exposure to high levels of a stressor are inhibitory, whereas low doses are stimulatory [133]. Such effects have been considered to be an example of evolutionary fitness [134] whereby organisms undergo metabolic adaptations in response to the interactions between the diverse stressors they are exposed to. Indeed, it has been postulated that such responses may be driven by antioxidants and oxidative stress [135]. Chronic stress on the colony has an important influence on colony dynamics [136], and if it surpasses a threshold, there is a risk of a destructive response, in this case, derived from the less efficient foraging of young bees [137]. This imbalance in the colony can drive the queen towards the limits of her egg-laying capacity [138], which may be accelerated by the interaction of several stressors. This behavior is similar to the Allee effect, which is related to the extinction of populations [139,140,141].

In terms of neonicotinoids, a significant increase in capped bee brood has been documented recently in small nucleus colonies exposed to low concentrations of chlothianidin over a period of 7 weeks [126]. It was reported that colonies needed to produce 1.57-fold more larvae to maintain a stable population when exposed to a concentration of 1 µg chlothianidin/L syrup, and that this effort increases as does exposure. Alternatively, effects that compensate nosemosis C [59] were evident in colonies during phase 2 of the disease, but if this does not control the stress provoked by the parasite in phase 3, it can eventually lead to colony collapse. Under the conditions of our field study, *N. ceranae* infection developed similarly over time in the FB and IB populations in both apiaries, with a relative infection below 5% at the end of the exposure period. The infection in the colonies had evolved to phase 2 (replacement) of nosemosis C by the end of the summer season [59]. Similar behavior has been described in colonies affected by varroosis, which might fail to survive because of their effort to redress the disturbed homeostasis [81]. Nevertheless, the pressure exerted by *V. destructor* was sufficiently well-controlled in the present study through the mandatory autumn treatment. In fact, according to the Spanish Ministry of Agriculture’s monitoring program, the prevalence of *V. destructor* was <1% in more than 80% of the colonies sampled in Castilla la Mancha in autumn 2014 and spring 2015 [142].

Immune defenses are costly to the individual, entering into a trade-off with other life-history traits such as reproduction, growth, and self-maintenance [134,143]. Suppression of the immune system may be favored by the interaction of several stressors, such as *N. ceranae* and TMX [77,144], and by high environmental temperatures [145], which could explain the prevalence of the BQCV virus in the colonies throughout the study. Significant increases in DWV loads have been documented in honey bees exposed to clothianidin, the metabolite of TMX, which has been related to inhibition of dorsal-1a gene transcription [36]. However, this increment was not observed here, with an increase in DWV only apparent in the autumn. These results are consistent with previous data [146] and they may be related to potential routes of transmission other than the main one through *Varroa* mite [147,148]. Moreover, the high environmental temperature in the spring and summer of 2015, together with a possible increase in thermogenesis due to exposure to TMX, might influence the inhibition of DWV reproduction, in part due to the host synthesis of heat shock proteins [149,150,151,152,153] and possibly by exceeding the optimal growth temperature range of this virus [154]. Nevertheless, further research is needed to understand the influence of high temperatures on the complex relationships between different traits of honey bee colonies and their pathogens.

Hormetic effects may not be statistically significant at the field level, in part because of their low magnitude and in part because of inadequate replication in experiments [155]. However, the difference between statistical significance and the biological relevance of an effect are not necessarily linked. Significance is a statistical measure reflecting whether an observed effect is likely to have occurred by chance alone [156]. EPPO standards (170) mean that statistical analysis may not be feasible in field studies due to the inherent variability of the end-points assessed and the limitations on replication [82]. Based on a Cohen’s d of 0.310, the lowest of our experimental results, it would be necessary to analyze 99 colonies/treatment to increase the statistical power up to 80%. Considering the possibility of establishing 8 hives/plot, this means a total of 12–13 plots per treatment. With this design, and according to the analysis carried out recently by the European Food Safety Authority [157], a significant effect between 7–8% could be detected. Few field studies in the literature follow such an exhaustive study design and they have generally failed to find deleterious effects at a colony level as a result of the use of neonicotinoids under good agricultural practices [158,159,160,161,162,163]. However, an increase in colony size is not always found as a result of neonicotinoid use, as observed here, suggesting that environmental conditions could play a key role in colony behavior.

## 5. Conclusions

Hormesis is common in arthropods and should be integrated into the dose response model as a continuum. However, the cost of the compensatory behavior adopted by the honey bee colonies over the long term is difficult to assess from an ecological point of view due to the complex relationships between the different traits affected. Further research is needed to understand the mechanisms underlining these compensation events, and the influence of environmental conditions on them, in order to define more appropriate management programs for this key pollinator.

## Figures and Tables

**Figure 1 insects-13-00371-f001:**
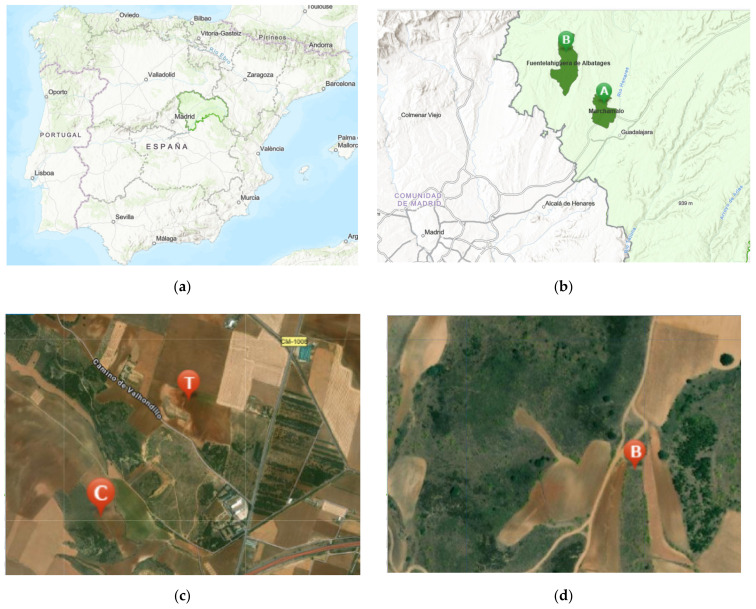
Site of the experimental study: (**a**) Map of Spain with the Province of Guadalajara highlighted in green; (**b**) Location of the apiaries during (A: Marchamalo) and after (B: Fuentelahiguera de Albatages) the exposure phase; (**c**) Detailed location of the apiaries during the exposure period in Marchamalo (C: control, T: treatment); (**d**) Detailed location of the apiaries after the exposure period in Fuentelahiguera de Albatages (B).

**Figure 2 insects-13-00371-f002:**
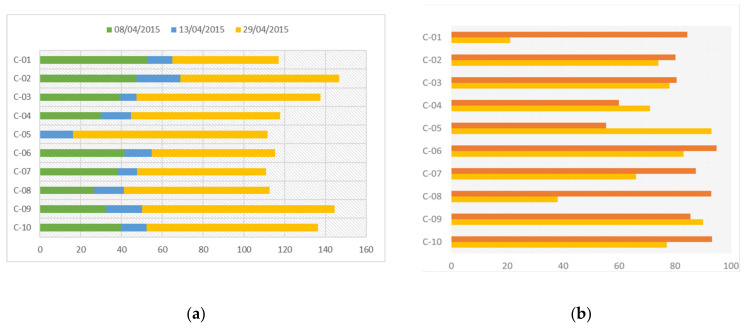
Melissopalynological analysis: (**a**) The % (*w*/*w*) *Brassica* t. loads in corbicular pollen at the beginning (8 April 2015), middle (13 April 2015), and end (29 April 2015) of winter oilseed rape (wOSR) flowering; (**b**) The frequency (%) of *Brassica* t. pollen in beebread 
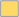
 and honey 
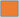
: C-01 to C-05 = Control hives; C-06 to C-10 = hives exposed to TMX.

**Figure 3 insects-13-00371-f003:**
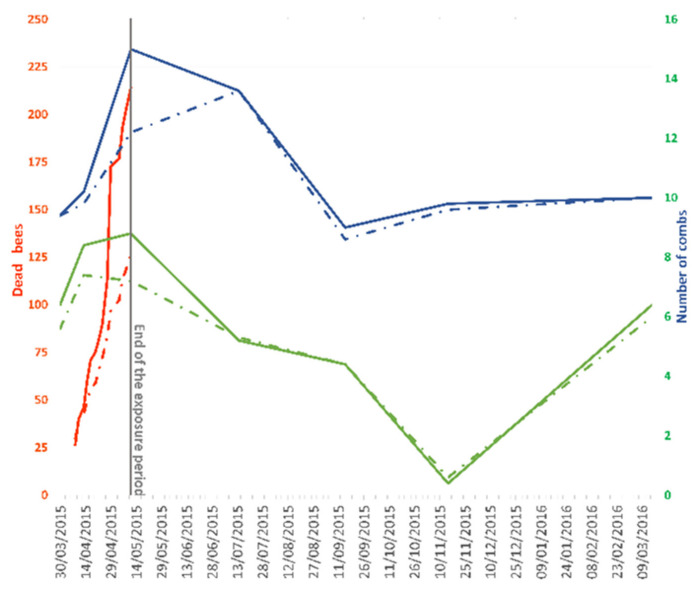
Colony strength parameters throughout the study in control and treatment (TMX) apiaries: mean accumulated dead bees at the entrance of the colonies (**— •****—** control; **—** TMX) and mean number of combs covered with bees (**— • ****—** control;**—**TMX) or brood (**— • ****—** control;**—** TMX).

**Figure 4 insects-13-00371-f004:**
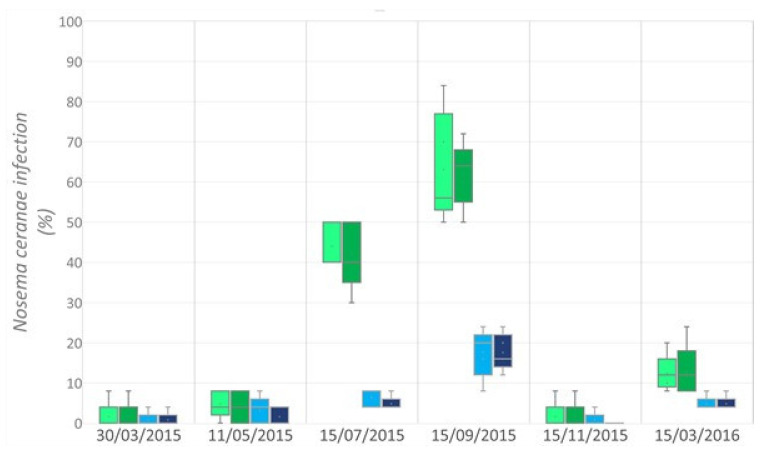
Box and Whisker plots of *N. ceranae* infection in non-forager honey bees (IB; 

 control; 
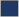
 TMX) and forager honey bees (FB; 
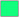
 control; 

 TMX).

**Table 1 insects-13-00371-t001:** Field schedule.

Sampling Date	Forager Death	Colony Strength	Pathogens	Bee Pollen	Beebread	Honey
30 March 2015	X	X	X			
8 April 2015	X			X		
10 April 2015	X					
13 April 2015	X	X		X		
15 April 2015	X					
17 April 2015	X					
20 April 2015	X					
22 April 2015	X					
24 April 2015	X					
27 April 2015	X					
29 April 2015	X			X		
4 May 2015	X					
6 May 2015	X					
8 May 2015	X					
11 May 2015	X	X	X		X	X
15 July 2015		X	X			
15 September 2015		X	X			
17 September 2015						
15 November 2015		X	X			
15 March 2016		X	X			

**Table 2 insects-13-00371-t002:** Descriptive statistics of the variables of strength in control and treatment (TMX) apiaries at the end of the exposure period.

Variable	Descriptive Statistics	Control (*n* = 5)	TMX (*n* = 5)
Accumulated deaths	Min	80	92
	Max	192	394
	Median	129	214
	Mean	127.6	214.4
	Standard deviation (SD)	44.106	116.468
	Variation coefficient (%)	34.565	54.323
	Mean 95% confidence interval	[72.836; 182.364]	[69.785; 359.015]
	Power (1-β)	0.379	
	Cohen’s d	0.863	
Number of combs with worker bees	Min	11	11
	Max	14	16
	Median	12	16
	Mean	12.2	15
	Standard deviation (SD)	1.095	2.236
	Mean 95% confidence interval	[10.84; 13.56]	[12.22; 17.77]
	Variation coefficient (%)	8.979	14.907
	Power (1-β)	0.541	
	Cohen’s d	1.245	
Number of combs with brood	Min	2	9
	Max	13	13
	Median	10.5	11.0
	Mean	9.0	11.0
	Standard deviation (SD)	4.966	1.826
	Variation coefficient (%)	55.184	16.598
	Mean 95% confidence interval	[1.097; 16.903]	[8.095; 13.905]
	Power (1-β)	0.103	
	Cohen’s d	0.310	
Honey production (kg)	Min	10.24	13.02
	Max	45.22	45.58
	Median	16.204	32.034
	Mean	24.755	31.977
	Standard deviation (SD)	15.075	12.21
	Variation coefficient (%)	60.894	38.184
	Mean 95% confidence interval	[6.038; 43.473]	[16.816; 47.137]
	Power (1-β)	0.165	
	Cohen’s d	0.475	

**Table 3 insects-13-00371-t003:** Detections of Black queen cell virus (BQCV) and Deformed wing virus (DWV) during the study in non-forager (IB) and forager bees (FB).

Hive	30 March 2015		11 May 2015		15 July 2015		15 September 2015		17 November 2015		15 March 2016	
	IB	FB	IB	FB	IB	FB	IB	FB	IB	FB	IB	FB
C 01	BQCV	-	BQCV	BQCV	BQCV	BQCV	-	-	DWV	DWV	DWV	DWV
C 02	BQCV	-	BQCV	-	BQCV	BQCV	BQCV	-	BQCV DWV	DWV	-	-
C 03	BQCV	BQCV-DWV	BQCV	BQCV	BQCV	BQCV	-	BQCV	BQCV-DWV	DWV	-	-
C 04	BQCV	BQCV-DWV	BQCV	DWV	BQCV	BQCV	BQCV	BQCV	BQCV-DWV	DWV	BQCV	-
C 05	DWV	BQCV	BQCV-DWV	-	DWV	BQCV	BQCV	-	BQCV-DWV	DWV	BQCV	-
C 06	BQCV	BQCV	BQCV	-	BQCV	BQCV	-	-	BQCV-DWV	DWV	BQCV	-
C 07	BQCV	-	BQCV	-	BQCV	BQCV	-	BQCV	BQCV-DWV	DWV	BQCV	-
C 08	BQCV	-	BQCV	BQCV	BQCV	BQCV	-	-	BQCV-DWV	DWV	-	DWV
C 09	-	BQCV	-	-	BQCV	BQCV	BQCV	-	BQCV-DWV	DWV	-	-
C 10	-	BQCV-DWV	-	BQCV	BQCV	BQCV	BQCV	-	BQCV-DWV	DWV	-	-

## Data Availability

The data presented in this study are available on reasonable request from the corresponding author.

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
