# Peer review of "Effects of Thiamethoxam-Dressed Oilseed Rape Seeds and *Nosema ceranae* on Colonies of *Apis mellifera iberiensis*, L. under Field Conditions of Central Spain. Is Hormesis Playing a Role?"

_insects, 2022, doi:10.3390/insects13040371_

Round 1
Reviewer 1 Report
Alonso-Prados et al. – TMX and hormesis in bees / Insects
This study is very appreciated even if it raises more questions than it answers. I am very supportive of publishing this research since there are very few studies that associate the current use of neonics and its prevalence in the ecosystem and hormetic effects on various living beings. It is also very thoughtful of the authors to invoke the Allee effect!
I have a few suggestions which I wish the authors could carry out in their manuscript or answer to make this manuscript even better, if published.
I believe the introduction has way more information in it which might not be very relevant to this study per se. One such example would be the EU regulatory status.
In general, while the study delineates between the foragers and the "interior bees", not much has been discussed about the drone bees. Are they considered "interior" as well? I assume that the drone numbers were so few as well as the colony mite levels are reasonable the drone counts could be ignored.
Line 101 - On the same subject, "interior bee" is not common terminology. Essentially, the non-foraging bees are primarily nurse bees with the exception of the drone and the queen bee in a hive. Hence, I would advise the authors to use terminologies such as foragers and non-foragers or workers and nurse bees. I do notice that the authors have noted the phenomenon of age polyethism in Line 104.
Results do seem to indicate that some hive parameters such as the honey production and number of combs and workers are marginally better than in the control. Do the authors have any data on differential productivity of the crop itself. Are the crop from the treated seeds more productive yielding more nectar and pollen and hence the treatment variation?
Sentence in Lines 115-116 needs a reference to show midgut damage by Nosema.
Line 149: "sampled and characterized"... Please provide more details. If by characterization it means the description in section 2.2, then the question would be, have the authors assessed the pesticide residue in the soil as well? Is it relevant to this study?
Table 1 - If the date format could be converted from DD/MM/YY to DD-MON-YYYY, it would be easier to comprehend for the readers from nations that use MDY formats.
Line 222,330 - What does 't' in "Brassica t., Prunus t., Cistus t., etc." refer to? Is it taxa? Please check for all unexplained abbreviations through the document.
Section 2.5 must be rewritten. The protocol is described 'in brief' but in a confusing manner at best. For example, the sentence in lines 240-242 - sounds as if dry MgSO4, NaCL and TSC were added to the homogenate. Expand the abbreviations PSA as well as C18.
Please provide details such as the manufacturer of the HPLC, mass spec, etc. Most of the chemicals' manufacturers as well as quality must be presented.
Line 250 - QuEChERS procedure -- normal readers and many students would not comprehend what this means.
Fig. 3 - Legend title - Please do avoid the use of the term 'Evolution'. Instead just say, "Colony parameters throughout the study..."
Line 436 - 'Allee's effect' should be 'the Allee effect'. The term is always preceded by an article.
References: The scientific names in the reference should be italicized; However, please check the journal formatting strictures.
Ref. 83 is in capital case. Check the author list for Ref.97. A number of references are not formatted properly.
Author Response
COMMENT 1.1: This study is very appreciated even if it raises more questions than it answers.
I am very supportive of publishing this research since there are very few studies that associate the current use of neonics and its prevalence in the ecosystem and hormetic effects on various living beings.
It is also very thoughtful of the authors to invoke the Allee effect!
I have a few suggestions which I wish the authors could carry out in their manuscript or answer to make this manuscript even better, if published.
ANSWER 1.1: Thanks a lot for your comment.
We very much appreciated the Reviewer effort in reviewing again our manuscript. We have considered all his/her comments, and we have performed some changes in our manuscript according to them.
COMMENT 1.2: I believe the introduction has way more information in it which might not be very relevant to this study per se. One such example would be the EU regulatory status.
ANSWER 1.2: The introduction has been reduced and reorganised, accordingly.
COMMNET 1.3: In general, while the study delineates between the foragers and the "interior bees", not much has been discussed about the drone bees. Are they considered "interior" as well? I assume that the drone numbers were so few as well as the colony mite levels are reasonable the drone counts could be ignored.
ANSWER 1.3: Yes, they are considered interior bees. Although the role of the drone on the dispersal of Varroa mites has been documented in the open literature Please consider the study was conducted between 2014 and 2016. In this time the levels of Varroa infection could be controlled in Spain with the compulsory autumn treatment stated in the national beekeeping program, as it is mentioned in the discussion.
COMMENT 1.4: Line 101 - On the same subject, "interior bee" is not common terminology. Essentially, the non-foraging bees are primarily nurse bees with the exception of the drone and the queen bee in a hive. Hence, I would advise the authors to use terminologies such as foragers and non-foragers or workers and nurse bees. I do notice that the authors have noted the phenomenon of age polyethism in Line 104
ANSWER 1.4: The text has been corrected accordingly considering foragers (FB) and no-foragers (IB) However, the abbreviation originally used is kept with harmonization purposes with the terminology used by our group in earlier papers on Nosema ceranae
COMMNET 1.5: Results do seem to indicate that some hive parameters such as the honey production and number of combs and workers are marginally better than in the control. Do the authors have any data on differential productivity of the crop itself. Are the crop from the treated seeds more productive yielding more nectar and pollen and hence the treatment variation?
ANSWER 1.5: As mentioned in section 2.1 the variety of OSR used in the experiment was Ginfizz variety, in both plots.
We do not have information on the production of nectar and pollen but only on seed production. The mean data of the seed production yield was better in the treated plot (4.2 PMG) than in the control plot (3.9 PMG) due to the protection against sucking insects derived from TMX treatment.
As mentioned in the discussion, The reason the production of OSR honey was higher in the apiary located in front of the treated plot may be related with the exposure to stressors (pathogens and chemicals ), which provoked the loss of forager bees was higher in the treated plot and this can have accelerated behavioral changes aimed at increasing food gathering in order to maintain the energy balance within the colony
COMMNENT 1.6: Sentence in Lines 115-116 needs a reference to show midgut damage by Nosema
ANSWER 1.6: Reference included
Dussaubat, C.; Brunet, J.-L.; Higes, M.; Colbourne, J.K.; Lopez, J.; Choi, J.-H.; Martín-Hernández, R.; Botías, C.; Cousin, M.; McDonnell, C.; et al. Gut pathology and responses to the microsporidium Nosema ceranae in the honey bee Apis mellifera. PloS one 2012, 7, e37017-e37017, doi:10.1371/journal.pone.0037017
COMMENT 1.7: Line 149: "sampled and characterized"... Please provide more details. If by characterization it means the description in section 2.2, then the question would be, have the authors assessed the pesticide residue in the soil as well? Is it relevant to this study?
ANSWER 1.7: Text amended in the ms as follows : the soil of the plots was sampled and characterized as described in section 2.2.
The soils samples were no analysed for pesticide residue
The text included in the discussion tries to give some explanation on the experimental results found in pollen and nectar residue levels based on the phys chem properties of the test item and the characteristics of the soils.
COMMENT 1.8: Table 1 - If the date format could be converted from DD/MM/YY to DD-MON-YYYY, it would be easier to comprehend for the readers from nations that use MDY formats.
ANSWER 1.8: Amended
COMMENT 1.9: Line 222,330 - What does 't' in "Brassica t., Prunus t., Cistus t., etc." refer to? Is it taxa? Please check for all unexplained abbreviations through the document.
ANSWER 1.9: This terminology is common in palynology studies and refers to “type” as explained in section2.4. Abbrevations were checked throughout the text.
COMMENT 1.10: Section 2.5 must be rewritten. The protocol is described 'in brief' but in a confusing manner at best. For example, the sentence in lines 240-242 - sounds as if dry MgSO4, NaCL and TSC were added to the homogenate. Expand the abbreviations PSA as well as C18
ANSWER 1.10: We have rewritten the entire subsection 2.5 paying special attention to all the comments raised by Reviewer 1. In addition, we have explained all the abbreviations at first mention, and we have also extended all the description of the different methodologies employed for each type of sample.
COMMENT 1.11: Please provide details such as the manufacturer of the HPLC, mass spec, etc. Most of the chemicals' manufacturers as well as quality must be presented.
ANSWER 1.11: We have included all the details requested by the Reviewer. It should be remarked that the specific data/details were only provided at first mention in order to avoid unnecessary repetitions.
COMMENT 1.12: Line 250 - QuEChERS procedure -- normal readers and many students would not comprehend what this means.
ANSWER 1.12: The acronym QuEChERS have been explained, and more information about the procedure has been included in the revised manuscript.
COMMENT 1.13: Fig. 3 - Legend title - Please do avoid the use of the term 'Evolution'. Instead just say, "Colony parameters throughout the study..."
ANSWER 1.13: Amended
COMMENT 1.14: Line 436 - 'Allee's effect' should be 'the Allee effect'. The term is always preceded by an article.
ANSWER 1.14: Amended
COMMENT 1.15: References: The scientific names in the reference should be italicized; However, please check the journal formatting strictures.
ANSWER 1.15: Checked and corrected
COMMENT 1.16: Ref. 83 is in capital case. Check the author list for Ref.97. A number of references are not formatted properly.
ANSWER 1.16: References checked .
Reviewer 2 Report
This is a thorough and well presented study. I have only minor changes to request:
lines 189 and 194: Please describe the type of dead bee trap and pollen trap used.
Fig 4: Is the vertical axis the % of sampled bees found to be infected? Do the bars show the range and the standard errors? The caption can be concise: "N. ceranae infection in ..."
line 396: "... and DWV was found ..."
Author Response
COMMENT 2.1: This is a thorough and well presented study. I have only minor changes to request.
ANSWER 2.1: Thanks a lot for your comment
We very much appreciated the Reviewer effort in reviewing again our manuscript. We have considered all his/her comments, and we have performed some changes in our manuscript according to them.
COMMENT 2.2: lines 189 and 194: Please describe the type of dead bee trap and pollen trap used.
ANSWER 2.2: Information included in the ms We used the box dead bee traps
COMMENT 2.3: Fig 4: Is the vertical axis the % of sampled bees found to be infected? Do the bars show the range and the standard errors? The caption can be concise: "N. ceranae infection in ..."
ANSWER 2.3: Yes it is. The nomenclature used is the same than the one used in previous papers of our group on Nosema ceranae. The representation is a box and whisker chart. Caption amended.
COMMENT 2.4: line 396: "... and DWV was found ..."
ANSWER 2.4: Corrected
Reviewer 3 Report
I believe that the authors can increase significantly the value of their manuscript by revising the text of manuscript.
- In general, the introduction is poorly documented. It doesn’t provide sufficient background information especially on the specific context of this work.
- The discussion is too speculative. The authors don’t explain and discuss deeply their results and indicate clearly the specific and the general scientific benefits.
- The conclusion is too general. The authors should develop a direct conclusion in relation to the question of this paper.
Author Response
COMMENT 3.1: I believe that the authors can increase significantly the value of their manuscript by revising the text of manuscript
ANSWER 3.1: We very much appreciated the Reviewer effort in reviewing again our manuscript. We have considered all his/her comments, and we have performed some changes in our manuscript according to them.
COMMENT 3.2: In general, the introduction is poorly documented. It doesn’t provide sufficient background information especially on the specific context of this work
ANSWER 3.2: We are sorry to see this comment . the results of the present study show the possible reaction of the colonies to chronic stressor as pathogens together with the exposure to thiamethoxam (TMX). The results show that under the experimental conditions, there is a response of the apiary exposed to TMX to keep the homeostasis of the colonies by increasing the honey bee population.
The introduction has been reduced and reorganised . It gives information on the effects of neonics and pathogens on honey bees, the possible synergic effects of these stressors and the difficulty on detect these interactions at field scale.
COMMENT 3.3: The discussion is too speculative. The authors don’t explain and discuss deeply their results and indicate clearly the specific and the general scientific benefits.
ANSWER 3.3:
We are sorry to see this comment Not always a research necessarily have to give benefits but raise questions and put the eye in some aspects which may be relevant to advance in the understanding of some processes in order to improve risk assessment and risk management procedures
In the discussion section, we firstly explain the exposure was guaranteed by a) melissopalinology , b) chemical analysis and c) dead bees found in the apiary in front of the treated plot . We recognise the exposure may have limited by:
1.- the interval between the application and exposure
- the environmental conditions during autumn and winter
3 the possible dilution of the residue inhive.
Despite this, the exposure was enough to observe an effect on the apiary in front of the treated plot regarding to the increment of dead foragers, production of honey and bee population. We consider the loss of forager bees can accelerate behavioral changes aimed at increasing food gathering in order to maintain the energy balance within the colony These changes bring about an increase in bee brood to compensate the interactions within the colony. This kind of behaviour is a hormetic like effect. Hormesis is common in arthropods. Indeed, it is likely that, given the many different processes involved, it cannot be characterized adequately by a single general mechanism and it is difficult to be statistically significant
To support our hypothesis we give examples cited in the literature on how - neonics, nosemosis C, Varroa, Viruses and high temperature can work as stressor provoking a reaction at colony level as the one observed in our experiment
Secondly, We discuss our results on the presence of BQCV . It can be related to immunodepression provoked by the interaction of stressors. The limited presence of DWV to autumn may be related to possible effects of the temperature.
We finally put our results on context with other filed studies where neonics has been tested under good agricultural practice and the experimental limitation to increase the robustness of the statistical analysis.
In order to make the reading easier , we have removed the paragraph regarding the effect of high temperature on the response threshold of the colony to sucrose demand (lines-469-479)
COMMENT 3.4: The conclusion is too general. The authors should develop a direct conclusion in relation to the question of this paper.
ANSWER 3.4: The conclusion has been rewritten as follows:
Hormesis is common in arthropods and should be integrated into the dose response model as a continuum. We consider necessary to characterize it and identify under which conditions these compensatory events appear in order to understand the dose response as a whole in order to define appropriate management programmes.